# Chemical Composition, Antioxidant, In Vitro and In Situ Antimicrobial, Antibiofilm, and Anti-Insect Activity of *Cedar atlantica* Essential Oil

**DOI:** 10.3390/plants11030358

**Published:** 2022-01-28

**Authors:** Miroslava Kačániová, Lucia Galovičová, Veronika Valková, Hana Ďuranová, Jana Štefániková, Natália Čmiková, Milena Vukic, Nenad L. Vukovic, Przemysław Łukasz Kowalczewski

**Affiliations:** 1Institute of Horticulture, Faculty of Horticulture and Landscape Engineering, Slovak University of Agriculture, Tr. A. Hlinku 2, 94976 Nitra, Slovakia; l.galovicova95@gmail.com (L.G.); veronika.valkova@uniag.sk (V.V.); n.cmikova@gmail.com (N.Č.); 2Department of Bioenergy, Food Technology and Microbiology, Institute of Food Technology and Nutrition, University of Rzeszow, 4 Zelwerowicza St., 35-601 Rzeszow, Poland; 3AgroBioTech Research Centre, Slovak University of Agriculture, Tr. A. Hlinku 2, 94976 Nitra, Slovakia; hana.duranova@uniag.sk (H.Ď.); jana.stefanikova@uniag.sk (J.Š.); 4Department of Chemistry, Faculty of Science, University of Kragujevac, 34000 Kragujevac, Serbia; milena.vukic@pmf.kg.ac.rs (M.V.); nvchem@yahoo.com (N.L.V.); 5Department of Food Technology of Plant Origin, Poznań University of Life Sciences, 31 Wojska Polskiego St., 60-624 Poznan, Poland

**Keywords:** atlas cedar, essential oil, antimicrobial activity, antibiofilm activity, repellent activity, vapor phase

## Abstract

The present study was designed to evaluate commercial cedar essential oil (CEO), obtained by hydrodistillation from cedar wood, in relationship to its chemical composition and antioxidant, *in vitro* and *in situ* antimicrobial, antibiofilm, and anti-insect activity. For these purposes, gas chromatography–mass spectrometry, DPPH radical-scavenging assay, agar and disc diffusion, and vapor phase methods were used. The results from the volatile profile determination showed that δ-cadinene (36.3%), (*Z*)-β-farnesene (13.8%), viridiflorol (7.3%), and himachala-2,4-diene (5.4%) were the major components of the EO chemical constitution. Based on the obtained results, a strong antioxidant effect (81.1%) of the CEO was found. CEO is characterized by diversified antimicrobial activity, and the zones of inhibition ranged from 7.33 to 21.36 mm in gram-positive and gram-negative bacteria, and from 5.44 to 13.67 mm in yeasts and fungi. The lowest values of minimal inhibition concentration (MIC) were noted against gram-positive *Micrococcus luteus* (7.46 µL/mL) and against yeast *Candida krusei* (9.46 µL/mL). It seems that the vapor phase of CEO can inhibit the growth of the microscopic filamentous fungi of the genus *Penicillium* according to *in situ* antifungal analysis on bread, carrots, and celery. This finding confirms the impact of CEO on the change in the protein structure of older biofilms of *Pseudomonas fluorescens* and *Salmonella enterica* subsp. *enterica*. Insecticidal activity of a vapor phase has also been demonstrated against *Pyrrhocoris apterus*. CEO showed various advantages on antimicrobial activity, and it is an ideal substitute for food safety.

## 1. Introduction

The use of plants in medicine was initiated many centuries ago, and secondary metabolites isolated from them, showing biological activity beneficial to health, are also used as fungicides in sustainable plant protection [1,2,3]. The essential oils (EO) are the volatile components responsible for the aroma of plants and are seeing an increasing interest from many industries due to their biological effect. Among the most frequently described distinctions are the antimicrobial [4,5,6,7], antioxidant [8], anti-inflammatory [9], anticancer [10], and antiparasitic [11] activities. EOs may even be used as fungicides in plant protection [12,13]. They also have an insect-repellent effect or inhibit egg laying and their growth; thus, they can also be used as natural insecticides [14,15,16]. Recent literature references describe the high antimicrobial efficacy of the EO vapor phase, in a relatively short time [17]. The principle of operation of the gas-phase technique is to generate EO vapors at a controlled temperature, creating an appropriate atmosphere and microenvironment [18].

Atlantic cedar (*Cedar atlantica*) is characterized by wide biological activity, including anti-inflammatory [19], anticancer [20,21], antioxidant [22], antimicrobial [23], and insecticidal [24] activities. An analgesic effect has also been described [25].

The antimicrobial and antibiofilm activities of many EOs against bacteria and fungi have been described [26,27], but particularly against *Candida* spp. [28]. For example, citronella and cinnamon EOs have been shown to have antibiofilm and anticandida properties [29]. Recently, it has been found that cascarilla bark and helichrysum oil strongly inhibited *Candida* biofilm and hyphal formation [30]. To date, the biological properties of cedar oil from Lebanese cedar (*Cedrus libani*) found in Lebanon and south-central Turkey have not been widely described. However, published data indicate that it exhibits antibacterial and antifungal activities [23,31,32].

Therefore, this paper is a report on the biological activity of *C. atlantica* essential oil (CEO). The CEO’s antioxidant as well as antimicrobial, antibiofilm, and insecticidal properties were analyzed. The chemical composition of the EO was also characterized. In addition, changes in the biofilm structure on glass and wood surfaces on *Salmonella enterica* subsp. *enterica* and *Pseudomonas fluorescens* were evaluated using MALDI-TOF MS Biotyper. The antibacterial and antifungal activities of the vapor phase of the essential oil in the food model was also analyzed. This is the first study where the chemical composition and antioxidant, antimicrobial, antibiofilm, and insecticidal activities of *C*. *atlantica* EO, commercially produced in Slovakia, were comprehensively determined.

## 2. Results

### 2.1. Chemical Composition of CEO

The analysis of the chemical composition of the CEO was performed using GC/MS and the identified compounds are presented in Table 1. A total of 31 compounds were identified, constituting 93.7% of all volatile substances. The obtained data revealed that the main component of the EO was δ-cadinene (36.35%), i.e., a mixture of (*Z*)-β-farnesene (13.8%) and β-himachalene (9.4%) followed by viridiflorol (7.3%) and himachala-2,4-diene (5.4%).

### 2.2. Antioxidant Activity of CEO

The antioxidant activity of CEO measured by the DPPH method was determined at 81.1 ± 0.4% of inhibition that corresponds to 825.00 ± 3.70 µg of TEAC/mL.

### 2.3. Antimicrobial Activity of CEO

In the analysis of CEO antimicrobial activity, gram-positive, gram-negative, biofilm-forming bacteria, and microscopic filamentous fungi were used. The CEO has been shown to vary in activity against the growth of the microorganisms used in the tests (Table 2). It can be seen that gram-positive (G^+^) bacteria with inhibition zones ranging from 15.23 ± 0.01 mm (*S. aureus*) to 21.36 ± 1.11 mm (*M. luteus*) were found to be more sensitive to CEO as compared to the gram-negative bacteria with inhibition zones varying between 9.86 ± 0.30 mm (*S. enterica*). The microscopic filamentous fungi range from 5.44 ± 0.15 mm (*P. aurantiogriseum*) to 13.67 ± 0.25 mm (*C. krusei*), respectively. Weak zones of inhibition (7.29 ± 1.26 mm, 9.36 ± 0.57 mm) were reported in the biofilm-forming bacteria, i.e., *P. fluorescens* and *S. enterica*, respectively. Then, the MIC test was performed, which showed the lowest MIC 50 (7.46 μL/mL) and MIC 90 (8.99 μL/mL) values for *M. luteus*, while the highest MIC 50 (28.48 μL/mL) and MIC 90 (32.16 µL/mL) values showed for *P. chrysogenum*. 

Our results showed that CEO (in all concentrations investigated) had a strong antibacterial effectiveness against the growth of *M. luteus* and *S. marcescens* on the bread model (Table 3). In effect, the concentrations exhibited similar inhibitory action (with no significant differences) on the growth of *M. luteus*, whereas the growth of *S. marcescens* was most inhibited by 125 µL/L of CEO. Regarding the carrot model (Table 4), significant inhibition of *M. luteus* growth was observed only after the application of 250 µL/L of CEO (Table 5). Interestingly, its remaining concentrations caused a three-fold increase in the growth of this bacterium indicating their probacterial activity. Weak probacterial activity of the EO in the concentration of 62.5 µL/L was also found against the growth of *S. marcescens*. On the other hand, the growth of the bacterium was moderately inhibited by 125 and 250 µL/L of CEO. On celery as a food model, CEO displayed moderate to weak (with increasing concentrations) antibacterial activity against *M. luteus* and a strong stimulating impact on the growth of *S. marcescens*.

Our results showed antifungal activity of CEO against *P. aurantiogriseum* (62.5, 125, and 250 µL/L), *P. expansum* (all tested concentrations), *P. chrysogenum* (250 and 500 µL/L), and *P. italicum* (62.5 and 500 µL/L) growing on the bread model (Table 6). On a carrot as a substrate (Table 7), the highest concentrations (250 and 500 µL/L) of the EO displayed very strong antifungal effectiveness against the growth of all fungi strains tested. Here, the lowest concentrations of the EO also had very strong antifungal activity against the growth of *P. aurantiogriseum* (62.5 and 125 µL/L), *P. expansum* (125 µL/L), and *P. chrysogenum* (125 µL/L). Very strong inhibitory activity of the EO against the growth of selected fungi strains was observed in *P. aurantiogriseum* (62.5 µL/L), *P. expansum* (250 µL/L), and *P. chrysogenum* (all tested concentrations) growing on celery as a food model (Table 8). Against *P. italicum*, a weak to moderate antifungal action was revealed but the growth of the fungi was weakly stimulated by the lowest concentration of the EO.

### 2.4. Antibiofilm Activity of CEO

The effect of CEO essential oil against the biofilm-producing bacterium, *P. fluorescens*, was evaluated by MALDI-TOF MS Biotyper mass spectrometry. The spectra of the control groups (planktonic cells and EO-untreated biofilm) developed in the same way (spectra not shown); the control planktonic cells were used as a control to compare the molecular changes of the biofilm.

Analysis of the mass spectra on days three and five of the microbial culture showed the same peaks, indicating the production of the same protein by young biofilms and control planktonic cells (Figure 1A,B), and no changes were observed in the bacterial cultures at the protein level. The differences were noticed only from the seventh day of biofilm growth on the analyzed surfaces, i.e., wood and glass (Figure 1C–F). Additionally, changes in the protein profile of the biofilm treated with CEO were visible. The analyzed essential oil influences the homeostasis of the bacterial biofilm formed on the examined surfaces.

Based on the constructed dendrogram (Dendrogram PF), showing the similarities in the biofilm structure in relation to the MSP distance, the shortest distance was found on the third and fifth days (PFG 3, PFW 3, PFG 5, PFW 5) in the planktonic (P) stage with the control groups and young biofilms (Figure 2). The short distances of MSP confirmed the similarity in the protein profile of the control groups. The short distances of MSPs, corresponding to the mass spectra, also had young biofilms and control plankton cells. Over time, the distance between the experimental groups of SMEs gradually increased. The mass spectra analyzed on days 12 and 14 of the experiment had the longest MSP distances, indicating changes in the molecular profile of *P. fluorescens*.

The effect of CEO against the biofilm-producing bacterium, *S. enterica*, was tested by MALDI-TOF MS Biotyper mass spectrometry. The spectra of the control groups (planktonic cells and EO-untreated biofilm) developed in the same way (spectra not shown); the control planktonic cells were used as a control to compare the molecular changes of the biofilm. The obtained mass spectra on days three and five of the culture showed similar peaks, indicating the same protein production by the young biofilms and control plankton cells (Figure 3A,B). At the protein level, no changes were observed in the bacterial cultures. The difference between the mass spectra of biofilms on the tested surfaces (wood and glass) and the control sample occurred from day seven (Figure 3C–F). As in the case of *P. fluorescens*, changes in the protein profile of the CEO-treated biofilm were also seen here.

Based on the constructed dendrogram (Dendrogram PF) showing the similarities in the biofilm structure in relation to the MSP distance, the shortest distance was found on the third and fifth days (SEG3, SEW3, SEG5, SEW5) in the planktonic (P) stage with the control groups and young biofilms (Figure 4). The short distances of MSP confirmed the similarity in the protein profile of the control groups. The short distances of MSPs, corresponding to the mass spectra, also had young biofilms and control plankton cells. Over time, the distance between the experimental groups of SMEs gradually increased. The mass spectra analyzed on day 14 of the experiment had the longest MSP distances, indicating changes in the molecular profile of *S. enterica*.

### 2.5. Anti-Insect Activity of CEO

As can be seen from the results presented in Table 9, all tested concentrations of the CEO gas phase showed insecticidal activity against *P. apterus*, the highest being observed at the concentration of 100%, causing the death of 100% of individuals. The concentration by half, i.e., 50%, killed 67% of the insects, and the concentration of 6.25% killed 20% of the individuals. 

## 3. Discussion

There were many studies focused on the analysis of cedar wood EO. The yields vary greatly in these studies, and this variability depended on the forest source as well as the used part of the tree. The variability in essential oil yield could be the consequence of many different factors such as the geographical origin and the time of harvest. The number of hours of light and sunshine is the main factor, which has an effect on photochemistry, and this fact has a significant effect on the accumulation of some compounds over a specific period of time due to the environmental conditions [33]. The essential oil production through secondary metabolism and the mechanism of production depend on the plant’s genetic constitution [34].

There is a higher yield of EOs extracted from tars than the yield of EO extracted from sawdust, but on the other side, this variability could be affected by pyrolysis process parameters, such as temperature, heating rate, and particle size on the product yield [35]. In our study, δ-cadinene (36.35%), (Z)-β-farnesene (13.8%), β-himachalene (9.4%), viridiflorol (7.3%) and himachala-2,4-diene (5.4%) were the major constituents of the total volatile components, respectively. 

There is a different concentration of β-himachalene (27.67–44.23%) in the EOs which were obtained from cedar wood (sawdust) (Itzer and Senoual forest) [36] in comparison with our study. Our results are not in line with the previous studies focused on EOs from Moroccan cedar wood [37,38,39] in which approximately forty compounds strongly dominated by α-himachalene (12.74%, 11.6%, 16.92%) and β-himachalene (33.45%, 33.8%, 43.18%) were found.

Many studies examined the chemical composition of cedar wood EO from different countries, such Lebanon [40], France [41], Morocco [39], and Algeria [42], and there were some significant chemical variations found in these studies.

In addition, there were also the antioxidant properties of the essential oils studied by using DPPH scavenging [36]. The results show that the essential oils from wood tar of *Cedrus atlantica* have strong antioxidant activity, similar to findings in our study.

Our results showed that CEO has different antimicrobial activities against the growth of microorganisms tested. The gram-positive (G^+^) bacteria *S. aureus* and *M. luteus* were found to be more sensitive to CEO as compared to the gram-negative bacteria *S. enterica*. Microscopic filamentous fungi ranged from 5.44 ± 0.15 mm (*P. aurantiogriseum*) to 13.67 ± 0.25 mm (*C. krusei*), respectively. Weak zones of inhibition (7.29 ± 1.26 mm, 9.36 ± 0.57 mm) were reported in the biofilm-forming bacteria, i.e., *P. fluorescens* and *S. enterica*, respectively. The lowest values for MIC 50 (7.46 μL/mL) and MIC 90 (8.99 μL/mL) were found for the *M. luteus*. By contrast, the highest MIC 50 (28.48 μL/mL) and MIC 90 (32.16 μL/mL) values were determined for *P. chrysogenum*. Derwich et al. (2010) revealed low to moderate antibacterial activity for *C. atlantica* leaf oil against a range of bacteria tested, with MIC values between 0.25 mg/mL and 1.62 mg/mL (MIC = 0.98 mg/mL for *Pseudomonas aeroginosa*, and MIC = 1.31 mg/mL for *Enterococcus faecalis*). Satrani et al. [43] also concluded that *C. atlantica* essential oil has antimicrobial activity against *Escherichia coli, B. subtilis*, *Micrococcus luteus*, and *Staphylococcus aureus*. A similar study demonstrated that the essential oils derived from Atlas cedar winged and wingless seeds were able to inhibit the growth of *Escherichia coli* at a concentration of 1/100 *v*/*v* [44]. *C. atlantica* essential oil in our study showed weak antifungal activity. The essential oil tested has fungistatic activity against almost all fungal strains studied. Thus, similar susceptibility levels were identified for *P. commune* and *A. niger*, with MIC values of 1% (*v*/*v*). *P. expansum*, *P. crustosum*, and *T. hyalocarpa* showed similar levels of susceptibility with MIC values of 0.5% (*v*/*v*) [45]. In a different study [23], the agar disc diffusion assay/aromatogram and minimum inhibitory concentration (MIC) were used for testing antimicrobial activities *in vitro* on gram-negative bacteria, *Escherichia coli* and *Salmonella*, as well as on gram-positive *Staphylococcus aureus*, *Bacillus subtilis*, and *Bacillus cereus.* It was found that the oil can be inactive against *Salmonella* and *S. aureus* with the agar disc diffusion assay. It was also found that *E. coli*, *B. subtilis,* and *B. cereus* are sensitive to cedarwood essential oil and showed effective bactericidal activity with a minimum inhibitory concentration of 0.4 μL/mL for *E. coli* and *B. cereus* and 0.2 μL/mL for *B. subtilis*. The results of the activity found strongly suggest that δ-cadinene is the principal compound responsible for the antimicrobial activity against *S. pneumoniae* [46]. It was observed that the aromatized himachalenes did not show antimicrobial activity against the gram-negative bacteria *B. cepacea*, *E. coli*, *K. pneumoniae,* and *P. aeruginosa.*

There are many cedar oils which are from the Cupressaceae family, mainly *huja occidentalis* and *Thuja plicata.* According to some studies [31], it seems that these oils have some antifungal activities against *C. albicans* and *Aspergillus niger.*

Our results showed that CEO had strong antibacterial effectiveness against the growth of *M. luteus* and *S. marcescens* on the bread, carrot, and celery models. The food industry primarily uses essential oils (EOs) as flavorings. Our results also showed antifungal activity of CEO against *P. aurantiogriseum*, *P. expansum*, *P. chrysogenum*, and *P. italicum* growing on the bread model. The highest concentrations of the EO displayed very strong antifungal effectiveness against the growth of all fungi strains tested on carrots and celery as substrates. EOs belong to the group of antimicrobial agents used as food preservatives for controlling food spoilage and foodborne pathogenic bacteria based on their growth inhibition properties [47]. The required volume of EOs to achieve complete inhibition via *in situ* tests was much higher than that with *in vitro* tests because the interaction between the EO and fat and protein contents of the bread may decrease the concentration of active volatile compounds [48]. The antimicrobial activity of some of the major compounds of CEO were studied. The antifungal activity of himachalene against *A*. *sydowii*, *A*. *parasiticus,* and *A*. *ochraceous* was also high (MIC 23.4–187.5 µg/mL), as reported against *A*. *fumigatus* for himachalol and other derivatives from *Cedrus* [49]. Antimicrobial activity of himachalene derivatives against gram-positive bacteria, *Bacillus subtilis*, *Micrococcus luteus,* and *Staphylococcus aureus*, and mycotoxigenic fungi, *Aspergillus parasiticus*, *A. ochraceous,* and *A. sydowii* were evaluated as well [50]. Viridiflorol has shown moderate antibacterial activity against *Mycobacterium tuberculosis*, the causative agent of tuberculosis, in an *in vitro* assay [51].

There are some benefits of using the vapor phase of Eos, such as the effect on final taste and aroma of the agricultural and food products or the regulation of their release [52]. It seems that according to our results there is a possibility to inhibit microbial growth using significantly lower concentrations due to the use of the vapor phase. The studies which are mentioned in this article have focused on the substantially higher concentrations by applying a gas phase or by direct contact so the efficiency should be higher, but on the other hand, there is still the need to find the lowest possible usable concentration. Mycotoxins are produced by fungal species, such as *Aspergillus*, *Penicillium*, and *Fusarium* in grain cereals, that colonize the plants in a field and can spread during the postharvest period [53]. The chronic problem, which still occurs, is the spoilage of stored food commodities. The consequences of the spoilage are qualitative and quantitative losses [54]. Essential oils with the most suitable use are the ones that have the widest antimicrobial effect against the real contamination of breads and vegetables by several strains. The future research of low-dose EOs mainly in agriculture and food production as well as their use as a part of organic production is important due to their ability for shelf-life extension. We found in our study that CEO was strongly effective against the growth of *M. luteus* and *S. marcescens* on the carrot and celery models, which is not in line with another study where no significant differences in inhibition between Gram-positive and Gram-negative bacteria in the vapor phase tests were found. Furthermore, *E. faecalis* (Gram-positive) was the most resistant to clove oil, whereas *S. choleraesuis* (Gram-negative) was the least inhibited by cinnamon oil. We also noticed some profungal effects on the bread and celery models against the growth of some *Penicillium* species. The effect was found to be cidal (inhibition percentage remains constant with time after removal of the antimicrobial atmosphere) for all the organisms except *Aspergillus flavus*. In this case, there was a reduction in percentage inhibition during the second week observed. It remained constant for the rest of the testing period after this reduction. There was a prolongation of the tests over 35 days to the control of the ability of EOs to provide the protection. There was no observed growth for any of the microorganisms except *A. flavus,* which confirms the static hypothesis for the latter organism. It is also interesting that estragol had no relevance in the antimicrobial effects of the tested EOs in the vapor phase, which proves that the basil seems to be totally ineffective [18].

Our vapor phase method should be adequate for the screening of large amounts of samples in a short time, and it should also be standardized for testing antimicrobial activity in the vapor phase due to the uniformity of the headspace.

The effect of CEO against the biofilm-producing bacteria, *P. fluorescens* and *S. enterica,* were evaluated by MALDI-TOF MS Biotyper mass spectrometry. Changes in the protein profile of both bacteria biofilm treated with CEO were visible. A biofilm can be defined as a complex matrix of microorganisms where the cells are bound together, and then they are attached to a biotic or abiotic surface [55,56]. The sticky gel is usually created by biofilms. This gel is composed of polysaccharides, proteins, and other organic components on a wet surface, and it was found in different environments including clinical and industrial ones, food processing environments, as well as drinking water distribution systems [57,58]. Bacteria in biofilm in comparison with planktonic cells in suspension seems to be more resistant to antibiotics and chemical agents [59,60]. The cedar wood oil in previous studies showed an inhibitory effect to fungal spores and vegetative cells [18], and thus, the CEO study was focused on determination of its antibiofilm activity. It was found out that the CEO dose dependently inhibits *C. albicans* biofilm formation, reducing biofilm formation by 87% at a concentration of 0.01% and completed inhibited biofilm formation at 0.1%. Therefore, the 0.01% concentration should be consider as an important concentration for further study [30]. The EO components showed a significant inhibitory effect on swimming, swarming, and twitching motility, EPS production, and biofilm formation of *P. fluorescens* [61]. The essential oils inhibited the formation of the biofilm of *S. enterica* and *S. typhimurium* in a different study [62]. The antibiofilm activity of the EOs against *Salmonella* was examined for the first time in our study. The antimicrobial and antibiofilm activities of some major compounds of these EOs [63] have been studied. One possible mechanism of the actions of EOs on biofilm should be connected with the ability of EOs to diffuse through the EPS matrix, allowing interactions with bacterial membrane proteins and decreasing the binding of planktonic cells to surfaces [64,65].

There are many studies focused on the insecticidal and repellent effects of plant extracts and essential oils against various stored-product pests [66,67,68,69,70]. The insecticidal activity was shown at all tested concentrations in the vapor phase. We recorded the killing of 100% of individuals at a concentration of 100%, the killing of 67% of individuals at a concentration of 50%, and 20% of individuals were killed at a concentration of 6.25%. There was acute toxicity and attraction–inhibitory activity of 28 commercially obtained essential oils and their major constituents were examined against the adults of *S. zeamais*. The greatest contact and fumigant toxicity amongst the tested essential oils was found in cinnamon oil [71]. Cinnamon oil and the most abundant constituent of the oil, trans-cinnamaldehyde, are known because of their insecticidal activity against several other coleopteran stored-product insects, including the rice weevil, *S. oryzae* L., *Chinese bruchid, Callosobruchus chinensis* L. [72], the red flour beetle, *Tribolium castaneum* Herbst [73], and the cigarette beetle, *Lasioderma serricorne* Fabricius [74]. The insecticidal principles, himachalol and ß-himachalene, showed potency against the pulse beetle and housefly [75].

## 4. Materials and Methods

### 4.1. Plant Material

*C. atlanticum* EO was obtained from Hanus s.r.o. (Nitra, Slovakia), which was prepared by hydrodistillation from cedar wood. The country origin of the cedar wood was Morocco. It was stored in the dark at 4 °C for the whole time that the analysis was carried out.

### 4.2. GC/MS of Essential Oil

The chemical characterization of CEO was performed in accordance with the method previously described by Kačániová et al. [76]. The GC/MS analysis of CEO was performed using Agilent 6890N gas chromatograph (Agilent Technologies, Santa Clara, CA, USA) coupled to quadrupole mass spectrometer 5975B (Agilent Technologies, Santa Clara, CA, USA).

### 4.3. DPPH Method

The antioxidant activity was measured using the method of Olszowy and Dawidowicz [77] in terms of the 2,2-diphenyl-1-picrylhydrazyl (DPPH) radical (Sigma Aldrich, Taufkirchen, Germany) with spectrophotometer Glomax (Promega Inc., Madison, WI, USA) at a wavelength of 515 nm. The percentage of inhibition was calculated as (A0−AA)/A0 × 100, where A0 was the absorbance of blank measurement, and AA was absorbance of sample. The antioxidant activity was expressed as antioxidant activity of Trolox related to 1 mL of sample (µg of TEAC/mL). Measurements were done in triplicate.

### 4.4. Microorganisms and Antimicrobial Activity

Gram-positive bacteria (*Bacillus subtilis* CCM 1999, *Enterococcus faecalis* CCM 4224, *Micrococcus luteus* CCM 732, *Staphylococcus aureus* subsp. *aureus* CCM 8223), Gram-negative bacteria (*Pseudomonas aeruginosa* CCM 3955, *Pseudomonas fluorescens* CCM 1969, *Salmonella enterica* subsp. *enterica* CCM 4420, *Serratia marcescens* CCM 8588), and yeasts (*Candida krusei* CCM 8271, *Candida albicans* CCM 8261, *Candida tropicalis* CCM 8223, *Candida glabrata* CCM 8270) were obtained from the Czech collection of micro-organisms (Brno, Czech Republic). The biofilm-forming bacteria, *Pseudomonas fluorescens*, was obtained from the fish, meat, and *Salmonella enterica* subsp. *enterica* from the chicken meat. Bacteria were identified with 16S rRNA sequencing and MALDI-TOF MS Biotyper.

Four strains of microscopic filamentous fungi of the genus *Penicillium* (*P. expansum, P. italicum, P. aurantiogriseum, P. chrysogenum*) were used to carry out the experiment. The fungi were isolated from a variety of materials usable in the food sector and were subsequently identified by ITS rDNA sequencing and MALDI-TOF MS Biotyper.

The disc diffusion method was used to determine the CEO’s antimicrobial activity. There was a 24-h inoculum for the bacteria on tryptone soy agar (Oxoid, Basingstoke, UK) at 37 °C, and the yeast on the Sabouraud dextrose agar (Oxoid, Basingstoke, UK) at 25 °C was adjusted to an optical density of 0.5 McFarland standard (1.5 × 10^8^ CFU/mL) and plated on Mueller Hinton agar plates (Oxoid, Basingstoke, UK). Sterile 6 mm discs soaked in 10 µL of CEO were placed on a layer of agar with a suspension of microorganisms and then incubated for another 24 h under the above-mentioned temperature conditions.

Cefoxitin (30 µg/disc), gentamicin (30 µg/disc) (Oxoid, Basingstoke, UK), and fluconazole (30 µg/disc) (Oxoid, Basingstoke, UK) were used as positive controls for Gram-negative and Gram-positive bacteria and yeast, respectively. The discs soaked with 0.1% DMSO (Centralchem, Bratislava, Slovakia) served as the negative control. Measurements were done in triplicate.

The inhibition zone above 15 mm was determined as very strong antimicrobial activity, the inhibition zone above 10 mm was determined as moderate activity, and the inhibition zone above 5 mm was determined as weak activity.

### 4.5. Minimal Inhibitory Concentration

The minimal inhibitory concentration (MIC) was determined in Mueller Hinton Broth (MHB, Oxoid, Basingstoke, UK) for bacteria and Sabouraud dextrose broth (Oxoid, Basingstoke, UK) for yeasts and fungi using the broth microdilution method in 96-well polystyrene microtiter plates according to Galovičová et al. [78]. Prior to the experiment, the bacteria, yeasts, and fungi were aerobically cultured for 24 h in MHB at 37 °C and SDB at 25 °C, respectively. As the negative and positive controls of maximal growth, the MHB, LGEO, and MHB with inoculum were employed, respectively.

### 4.6. Antimicrobial Analysis In Situ on a Food Model

All four fungal strains (*P. expansum*, *P. italicum*, *P. aurantiogriseum*, *P. chrysogenum*) and two bacterial strains, *M. luteus* and *S. marcescens*, were selected based on the results of *in vitro* antimicrobial activity and were used to estimate the antimicrobial activity of LGEO *in situ*. Here, three commonly consumed foods (wheat bread, carrots, celery) were used as substrates for the growth of selected microorganism species. The bakery food model was produced in the Laboratory of Cereal Technologies (Research Center AgroBioTech, SUA in Nitra) according to the methodology described in the study by Kačániová et al. [76]. The experiment itself was carried out according to Galovičová et al. [78]. After cooling, the slices of bread were cut to a thickness of 1.5 cm and placed in glass jars (Bormioli Rocco, Fidenza, Italy; 0.5 L). Before inoculation, the bacterial and fungal strains were cultured on Tryptone Soya agar (TSA, Oxoid, Basingstoke, UK) for 24 h at 37 °C and on SDA at 25 °C for five days, respectively. Consequently, the inoculum of tested strains was applied by stabbing with an injection pin three times on the bread substrate. Then, a sterile filter paper disc (6 cm) was placed under the jar top, and 100 µL of LGEO in concentrations of 62.5, 125, 250, and 500 µL/L (diluted in ethyl acetate) were applied to it. The control bread was not treated with LGEO. Finally, the hermetically closed jars were stored in an incubator for 7 (bacteria; 37 ± 1 °C) and 14 days (fungi; 25 ± 1 °C).

For vegetables (carrots and celery) as food models, the methodology was slightly modified. Firstly, MHA was poured into the bottom and lids of the Petri dishes (PD; 60 mm). Sliced carrots and celery (0.5 mm) were firstly placed on the agar poured at the bottom, and the inoculum was prepared as described for the bread model but only 10 µL of LGEO (in the same concentrations) was applied on the sterile filter paper disc and then placed at the pin of PD. Subsequently, PDs were hermetically closed and cultivated at 37 °C for 7 days and at 25 °C for 14 days for bacteria and fungi, respectively. Measurements were done in triplicate.

### 4.7. Analysis of Differences in Biofilm Development with MALDI-TOF MS Biotyper

The changes of protein spectra during biofilm development after the CEO addition were evaluated by MALDI-TOF MS Biotyper (Bruker Daltonics) according to the method described in details earlier [76]. *P. fluorescens* and *S. enterica* were used as the representative biofilm forming bacteria. The spectra were obtained by automated analysis and the same sample similarities were used to generate the global standard spectrum (MSP), and 19 MSPs were generated from the spectra by MALDI Biotyper 3.0 and grouped into dendrograms using Euclidean distance [79].

### 4.8. Anti-Insect Activity

The insecticidal activity of CEO was tested using *Pyrrhocoris apterus*. *P. apterus* were collected in a sterile glass container of *Hibiscus syriacus* aseptically in the locality of Nitra (48°18′27.47′′ N 18°05′4.31′′ E). The EO was diluted in 0.1% polysorbate solution. Concentrations of 100%, 50%, 25%, 12.5%, and 6.25% were tested. A 0.1% polysorbate was used as the negative control. A total of 30 individuals of *P. apterus* were placed in 90 mm Petri dishes (PD) with vents. A circle of sterile filter paper was placed into the PD lid. One hundred μL of the appropriate concentration of CEO was applied to the filter paper and the plates were sealed with parafilm. *P. apterus* were exposed to CEO vapors for 24 h at room temperature. After exposure, live and dead subjects were counted, and the percentage of insecticidal activity was calculated. Measurements were done in triplicate.

### 4.9. Statistical Data Evaluation

SAS^®^ software version 8 (SAS Institute, Cary, NC, USA) was used for data processing. The results of the MIC value (concentration that caused 50% and 90% inhibition in bacterial growth) were determined by logit analysis.

## 5. Conclusions

In the current study, chemical composition, antioxidant, antimicrobial (*in vitro*, *in situ*), and antibiofilm activities, as well as anti-insect activity of commercial CEO obtained from the Hanus Company in Slovakia were investigated. The major compounds of CEO tested were δ-cadinene (36.35%), (Z)-β-farnesene (13.8%), β-himachalene (9.4%), viridiflorol (7.3%), and himachala-2,4-diene (5.4%), and antioxidant activity was 81.1%. *In vitro* antibacterial activity showed that CEO was the most effective against the growth of *M. luteus* and antifungal activity against *C. krusei*, and the lowest values for MIC were determined for the same microorganisms. Considering the fungal strains, the concentration of ≥250 µL/L of the EO was found to inhibit the growth of all *Penicillium* spp. (*P. aurantiogriseum*, *P. expansum*, *P. chrysogenum*, and *P. italicum*). Additionally, it was found that CEO influenced the mass spectra of biofilms produced on various surfaces, as it was detected by MALDI-TOF MS Biotyper. Using a food model, *in situ* antimicrobial analysis showed an inhibitory effectiveness of CEO against the growth of bacteria (*M. luteus* and *S. marcescens* and fungi inoculated on bread, carrots, and celery as the growing substrates in the vapor phase. Very good anti-insect activity of CEO against *P. apterus* was evaluated.

## Figures and Tables

**Figure 1 plants-11-00358-f001:**
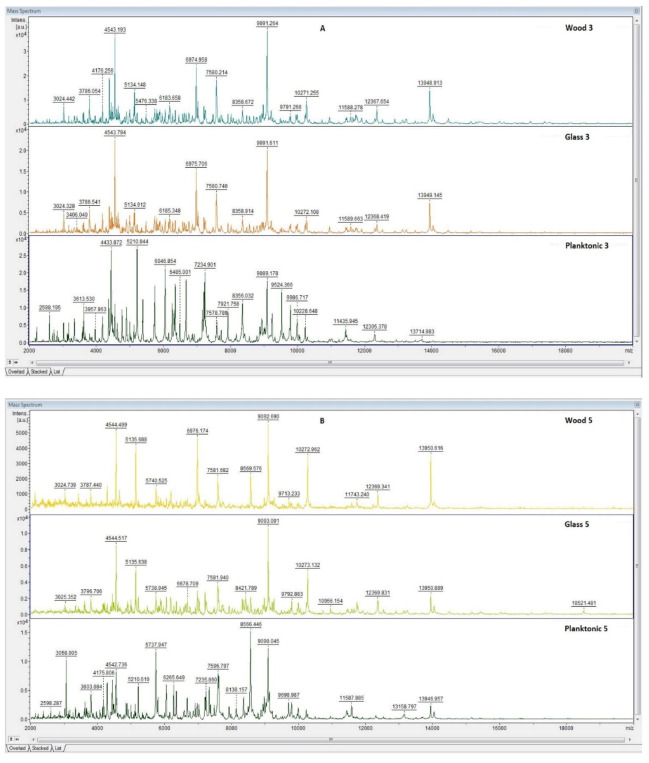
MALDI-TOF mass spectra of *P. fluorescens* biofilm during development after the addition of CEO: (**A**) 3rd day, (**B**) 5th day, (**C**) 7th day, (**D**) 9th day, (**E**) 12th day, and (**F**) 14th day.

**Figure 2 plants-11-00358-f002:**
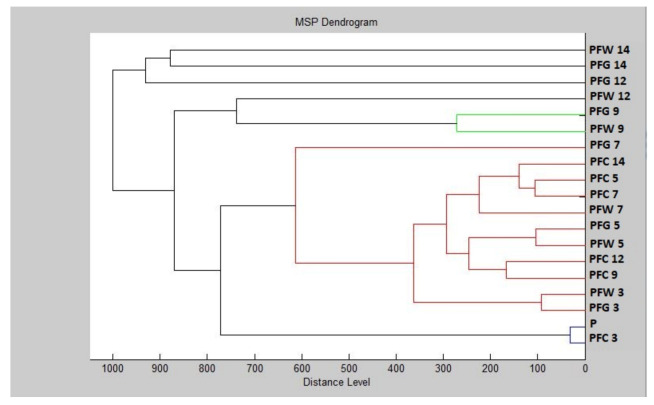
Dendrogram of *P. fluorescens* generated using MSPs of the planktonic cells and the control. PF, *P. fluorescens*; C, control; G, glass; W, wood; and P, planktonic cells.

**Figure 3 plants-11-00358-f003:**
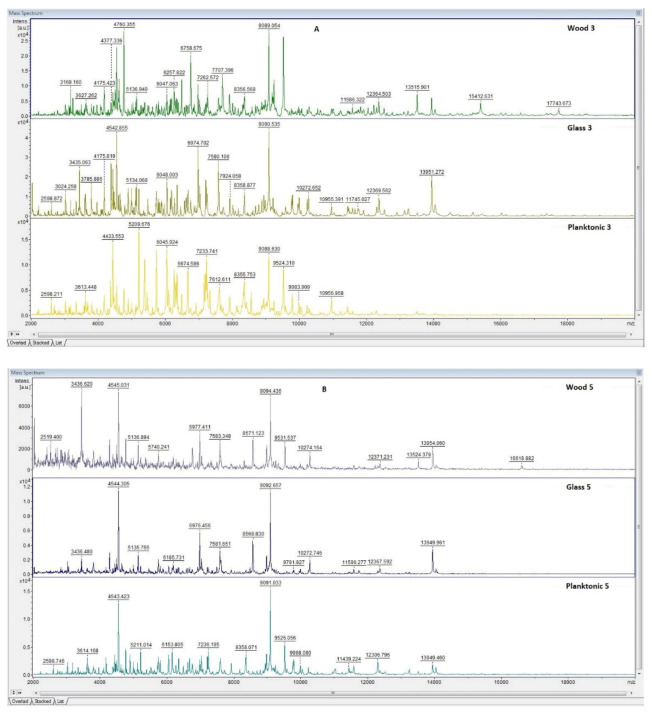
MALDI-TOF mass spectra of *S. enterica* biofilm during development after the addition of CEO: (**A**) 3rd day, (**B**) 5th day, (**C**) 7th day, (**D**) 9th day, (**E**) 12th day, and (**F**) 14th day.

**Figure 4 plants-11-00358-f004:**
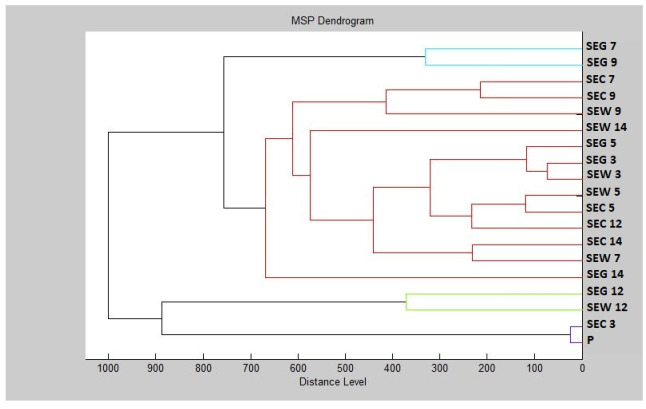
Dendrogram of *S. enterica* generated using MSPs of the planktonic cells and the control. SE, *S. enterica*; C, control; G, glass; W, wood; and P, planktonic cells.

**Table 1 plants-11-00358-t001:** Chemical composition of CEO.

No	RI ^a^	Compound ^b^	% ^c^
1	926	α-thujene	tr
2	938	α-pinene	tr
3	948	camphene	1.6
4	977	sabinene	0.1
5	980	β-pinene	0.2
6	992	β-myrcene	0.3
7	1023	p-cimene	0.2
8	1028	α-limonene	0.8
9	1033	1,8-cineole	0.9
10	1060	γ-terpinene	2.4
11	1189	α-terpineol	0.3
12	1331	neoisolongifolene	0.2
13	1371	isoledene	1.3
14	1391	isolongifolene	0.6
15	1419	α-cedrene	0.6
16	1429	himachala-2,4-diene	5.4
17	1439	γ-elemene	0.4
18	1443	(Z)-β-farnesene	13.8
19	1483	γ-himachalene	0.2
20	1521	β-himachalene	9.4
21	1525	δ-cadinene	36.3
22	1551	α-calacorene	3.0
26	1578	α-cedrene epoxide	1.5
27	1593	viridiflorol	7.3
28	1600	longiborneol	1.7
29	1601	widdrol	0.5
30	1643	epi-α-cadinol	2.2
31	1671	α-bisabolol	1.2
	total		93.7

^a^ Values of retention indices on HP-5MS column; ^b^ identified compounds; ^c^ tr—compounds identified in amounts less than 0.1%.

**Table 2 plants-11-00358-t002:** Antimicrobial activity of CEO.

Microorganism	Zone Inhibition (mm)	Activity of EO	MIC 50 (µL/mL)	MIC 90 (µL/mL)	ATB (30 µg/disc)
Gram-positive bacteria					
*Bacillus subtilis*	17.33 ± 0.01	***	12.23	35.41	27.00 ± 0.05
*Enterococcus faecalis*	18.65 ± 0.20	***	23.38	284.56	29.00 ± 0.08
*Micrococcus luteus*	21.36 ± 1.11	***	7.46	8.99	30.00 ± 0.02
*Staphylococcus aureus*	15.23 ± 0.01	***	12.15	24.33	28.00 ± 0.06
Gram-negative bacteria					
*Pseudomonas aeruginosa*	15.78 ± 0.67	***	15.36	23.38	28.00 ± 0.08
*Pseudomonas fluorescens*	11.22 ± 0.83	**	21.18	35.16	28.00 ± 0.06
*Salmonella enterica*	9.86 ± 0.30	*	18.48	24.25	27.00 ± 0.09
*Serratia marcescens*	8.27 ± 1.51	*	21.43	32.36	30.00 ± 0.04
*Psuedomonas fluorescens* biofilm	7.29 ± 1.26	*	31.82	52.44	27.00 ± 0.01
*Salmonella enterica* biofilm	9.36 ± 0.57	*	24.67	41.52	27.00 ± 0.02
Yeasts					
*Candida albicans*	10.12 ± 1.00	**	17.32	22.62	28.00 ± 0.06
*Candida glabrata*	10.44 ± 0.88	**	18.64	24.54	27.00 ± 0.05
*Candida krusei*	13.67 ± 0.25	**	9.46	12.53	26.00 ± 0.03
*Candida tropicalis*	12.36 ± 0.20	**	24.12	28.26	27.00 ± 0.08
Fungi					
*Penicillium aurantiogriseum*	5.44 ± 0.15	*	23.45	28.65	27.00 ± 0.06
*Penicillium expansum*	7.22 ± 0.55	*	22.15	26.35	26.00 ± 0.09
*Penicillium chrysogenum*	8.32 ± 0.20	*	28.48	32.16	28.00 ± 0.08
*Penicillium italicum*	7.36 ± 0.05	*	25.35	31.85	29.00 ± 0.09

Means ± SD, n = 3; * weak antimicrobial activity (zone 5–10 mm); ** moderate inhibitory activity (zone > 10 mm); *** very strong inhibitory activity (zone > 15 mm); ATB—antibiotics; positive control (cefoxitin for G^−^, gentamicin for G^+^, fluconazole for yeast and fungi).

**Table 3 plants-11-00358-t003:** *In situ* analysis of the antibacterial activity of the vapor phase of CEO on bread.

Bacteria	Bacterial Growth Inhibition [%] Bread			
	62.5 (µL/L)	125 (µL/L)	250 (µL/L)	500 (µL/L)
*Micrococcus luteus*	63.24 ± 5.21 ^a^	63.08 ± 3.88 ^a^	65.75 ± 4.65 ^a^	67.90 ± 4.36 ^a^
*Serratia marcescens*	66.70 ± 5.33 ^a^	87.34 ± 3.81 ^b^	73.93 ± 6.79 ^c^	40.18 ± 4.96 ^d^

Means ± standard deviation (n = 3). Values followed by different superscript within the same row are significantly different (*p* < 0.05).

**Table 4 plants-11-00358-t004:** *In situ* analysis of the antibacterial activity of the vapor phase of CEO on carrots.

Bacteria	Bacterial Growth Inhibition [%] Carrots			
	62.5 (µL/L)	125 (µL/L)	250 (µL/L)	500 (µL/L)
*Micrococcus luteus*	−308.56 ± 5.66 ^a^	−310.56 ± 7.12 ^a^	81.13 ± 4.28 ^b^	−302.56 ± 5.64 ^a^
*Serratia marcescens*	−15.65 ± 2.83 ^a^	33.79 ± 4.77 ^b^	31.50 ± 5.19 ^b^	−0.093 ± 5.76 ^c^

Means ± standard deviation (n = 3). Values followed by different superscript within the same row are significantly different (*p* < 0.05); the negative values indicate probacterial activity of the essential oil against the growth of bacteria strains.

**Table 5 plants-11-00358-t005:** *In situ* analysis of the antibacterial activity of the vapor phase of CEO on celery.

Bacteria	Bacterial Growth Inhibition [%] Celery			
	62.5 (µL/L)	125 (µL/L)	250 (µL/L)	500 (µL/L)
*Micrococcus luteus*	57.14 ± 1.23 ^a^	23.29 ± 0.99 ^b^	11.77 ± 2.56 ^c^	5.15 ± 2.09 ^d^
*Serratia marcescens*	−71.40 ± 3.20 ^a^	−83.40 ± 5.61 ^b^	−311.09 ± 6.75 ^c^	−318.00 ± 4.73 ^c^

Means ± standard deviation (n = 3). Values followed by different superscript within the same row are significantly different (*p* < 0.05); the negative values indicate probacterial activity of the essential oil against the growth of bacteria strains.

**Table 6 plants-11-00358-t006:** *In situ* analysis of the antifungal activity of the vapor phase of CEO on bread.

Fungi	Mycelial Growth Inhibition [%] Bread			
	62.5 (µL/L)	125 (µL/L)	250 (µL/L)	500 (µL/L)
*Penicillium aurantiogriseum*	25.52 ± 5.88 ^a^	27.81 ± 3.49 ^a^	30.83 ± 4.93 ^a^	−3.83 ± 4.62 ^b^
*Penicillium* *expansum*	20.04 ± 6.05 ^a^	16.84 ± 5.33 ^a^	20.27 ± 5.87 ^a^	41.85 ± 3.76 ^b^
*Penicillium* *chrysogenum*	2.31 ± 3.12 ^a^	2.1 ± 4.78 ^a^	33.02 ± 5.56 ^b^	21.86 ± 5.91 ^b^
*Penicillium* *italicum*	53.12 ± 6.93 ^a^	23.97 ± 3.21 ^b^	−1.16 ± 4.11 ^c^	44.70 ± 5.39 ^a^

Means ± standard deviation (n = 3). Values followed by different superscript within the same row are significantly different (*p* < 0.05); the negative values indicate profungal activity of the essential oil against the growth of fungi strains.

**Table 7 plants-11-00358-t007:** *In situ* analysis of the antifungal activity of the vapor phase of CEO on carrots.

Fungi	Mycelial Growth Inhibition [%] Carrots			
	62.5 (µL/L)	125 (µL/L)	250 (µL/L)	500 (µL/L)
*Penicillium aurantiogriseum*	100.00 ± 0.00 ^a^	98.72 ± 5.64 ^a^	100.00 ± 0.00 ^a^	100.00 ± 0.00 ^a^
*Penicillium expansum*	0.00 ± 0.00 ^a^	100.00 ± 0.00 ^b^	100.00 ± 0.00 ^b^	100.00 ± 0.00 ^b^
*Penicillium chrysogenum*	0.00 ± 0.00 ^a^	100.00 ± 0.00 ^b^	100.00 ± 0.00 ^b^	91.57 ± 6.24 ^b^
*Penicillium italicum*	0.00 ± 0.00 ^a^	1.21 ± 3.35 ^a^	100.00 ± 0.00 ^b^	100.00 ± 0.00 ^b^

Means ± standard deviation (n = 3). Values followed by different superscript within the same row are significantly different (*p* < 0.05).

**Table 8 plants-11-00358-t008:** *In situ* analyses of the antifungal activity of the vapor phase of CEO on celery.

Fungi	Mycelial Growth Inhibition [%] Celery			
	62.5 (µL/L)	125 (µL/L)	250 (µL/L)	500 (µL/L)
*Penicillium aurantiogriseum*	100.00 ± 0.00 ^a^	48.15 ± 4.76 ^b^	−9.08 ± 5.07 ^c^	−11.11 ± 4.48 ^c^
*Penicillium* *expansum*	36.11 ± 3.97 ^a^	−140.70 ± 6.67 ^b^	100.00 ± 0.00 ^c^	−26.22 ± 4.89 ^d^
*Penicillium* *chrysogenum*	97.78 ± 5.77 ^a^	98.39 ± 4.96 ^a^	95.08 ± 4.46 ^a^	94.74 ± 5.29 ^a^
*Penicillium* *italicum*	−6.10 ± 4.36 ^a^	16.35 ± 4.77 ^b^	37.86 ± 5.14 ^c^	25.00 ± 6.02 ^bc^

Means ± standard deviation (n = 3). Values followed by different superscript within the same row are significantly different (*p* < 0.05); the negative values indicate profungal activity of the essential oil against the growth of fungi strains.

**Table 9 plants-11-00358-t009:** Insecticidal activity of vapor phase of CEO against *P. apterus*.

Concentration [%]	Number of Living Individuals	Number of Dead Individuals	Insecticidal Activity [%]
100	0	30	100
50	10	20	67
25	15	15	50
12.5	20	10	33
6.25	25	5	20
Control group	30	0	0

## Data Availability

Data is contained within the article.

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
