# Peer review of "Chemical Composition, Antioxidant, In Vitro and In Situ Antimicrobial, Antibiofilm, and Anti-Insect Activity of Cedar atlantica Essential Oil"

_plants, 2022, doi:10.3390/plants11030358_

Round 1
Reviewer 1 Report
Dear Editor,
I carefully read the submission title 'Chemical Composition, Antioxidant, In Vitro and In Situ Antimicrobial, Antibiofilm and Anti-Insect Activity of Cedar atlantica Essential Oil'. In fact more recently there has been an increasing interest to MAP (Medicinal and Aromatic Plants).
My first impression that the paper contain new information and title of the manuscript cover its content. The summary is appropriate and the aim of the work clearly established. The methods are used are adequate and used sophisticated techniques and equipment's. I found the results very reliable. Discussion and conclusions are well documented and scientifically coherent.
However, I have some corrections and additions on it before acceptance.
ABSTRACT: The first sentence should be rewritten. ....Cedar atlantica belongs to plants known for their many biological properties.???
Do not abbreviate EO when first time used (Line 26). Maybe change as CEO.
Same MIC should be not abbreviated when first time used in abstract.
The full name of microorganisms used must be given when use first time such as M. luteus etc.
INTRODUCTION:
Plants have been used in medicine for centuries, and secondary metabolites isolated 41 from them, showing biological activity beneficial to health, are also used in the production 42 of drugs [1]. This sentence needs more references
I suggest below ones
Zia-Ul-Haq, M.; Ahmad, S.; Qayum, M.; Ercisli, S. Compositional studies and antioxidant potential of Albizia lebbeck (L.) Benth. Pods and seeds. Turk. J. Biol. 2013, 37, 25-32.
Mollova, S.; Fidan, H.; Antonova, D.; Bozhilov, D.; Stanev, S.; Kostova, I.; Stoyanova, A. Chemical composition and antimicrobial and antioxidant activity of Helichrysum italicum (Roth) G. Don subspecies essential oils. Turk. J. Agric. For, 2020, 44, 371-378.
please correct this sentence as....used even as fungicides in sustainable plant protection.
4.1. Essential oil must be Plant material
C. atlanticum EO was obtained from Hanus, s.r.o. (Nitra, Slovakia)??? Which company????
Author Response
Reviewer #1
I carefully read the submission title 'Chemical Composition, Antioxidant, In Vitro and In Situ Antimicrobial, Antibiofilm and Anti-Insect Activity of Cedar atlantica Essential Oil'. In fact more recently there has been an increasing interest to MAP (Medicinal and Aromatic Plants).
My first impression that the paper contain new information and title of the manuscript cover its content. The summary is appropriate and the aim of the work clearly established. The methods are used are adequate and used sophisticated techniques and equipment's. I found the results very reliable. Discussion and conclusions are well documented and scientifically coherent.
However, I have some corrections and additions on it before acceptance.
We would like to thank the review for the valuable comments.
Point 1: ABSTRACT: The first sentence should be rewritten. ....Cedar atlantica belongs to plants known for their many biological properties.???
Response: The first sentence was removed.
Point 2: Do not abbreviate EO when first time used (Line 26). Maybe change as CEO.
Response: Before abbreviation EO is cedar essential oil (CEO).
Point 3: Same MIC should be not abbreviated when first time used in abstract.
Response: It was changed.
Point 4: The full name of microorganisms used must be given when use first time such as M. luteus etc.
Response: It was changed.
Point 5: INTRODUCTION: Plants have been used in medicine for centuries, and secondary metabolites isolated 41 from them, showing biological activity beneficial to health, are also used in the production 42 of drugs [1]. This sentence needs more references
I suggest below ones
Zia-Ul-Haq, M.; Ahmad, S.; Qayum, M.; Ercisli, S. Compositional studies and antioxidant potential of Albizia lebbeck (L.) Benth. Pods and seeds. Turk. J. Biol. 2013, 37, 25-32.
Mollova, S.; Fidan, H.; Antonova, D.; Bozhilov, D.; Stanev, S.; Kostova, I.; Stoyanova, A. Chemical composition and antimicrobial and antioxidant activity of Helichrysum italicum (Roth) G. Don subspecies essential oils. Turk. J. Agric. For, 2020, 44, 371-378.
please correct this sentence as....used even as fungicides in sustainable plant protection.
Response: It was corrected.
Point 6: 4.1. Essential oil must be Plant material
Response: It was corrected.
Point 7: C. atlanticum EO was obtained from Hanus, s.r.o. (Nitra, Slovakia)??? Which company????
Response: It was corrected.
Reviewer 2 Report
Research questions are well defined and within the aims and the scope of the journal. The introduction is adequate and including in suitable way the relevant earlier publications. Materials should be better described. Methods are properly described and used in a way that is possible to replicate. The investigation is mainly performed to good technical standards. It is no ethical problem involved.
Suggestions:
Lines 30, (M. luteus), 31, 34. Write the correct name of genus, not only the first letter, when it is appearing for the first time in the text.
The same in Tables 2., 6., 7. And 8. Write the name of genus, not only the first letter to be more informative for the readers.
Line 364. Give in the details the data on the origin of cedar wood.
Author Response
Reviewer #2
Research questions are well defined and within the aims and the scope of the journal. The introduction is adequate and including in suitable way the relevant earlier publications. Materials should be better described. Methods are properly described and used in a way that is possible to replicate. The investigation is mainly performed to good technical standards. It is no ethical problem involved.
Suggestions:
We would like to thank the review for the valuable comments.
Point 1: Lines 30, (M. luteus), 31, 34. Write the correct name of genus, not only the first letter, when it is appearing for the first time in the text.
Response: It was corrected.
Point 2: The same in Tables 2., 6., 7. And 8. Write the name of genus, not only the first letter to be more informative for the readers.
Response: It was corrected.
Point 3: Line 364. Give in the details the data on the origin of cedar wood.
Response: It was added
Reviewer 3 Report
In the manuscript "Chemical Composition, Antioxidant, In Vitro and In Situ Antimicrobial, Antibiofilm and Anti-Insect Activity of Cedar atlantica Essential Oil" the authors examined the biological effects of Cedar atlantica EO. The work is very similar to their previous work "Chemical Composition, In Vitro and In Situ Antimicrobial and Antibiofilm Activities of Syzygium aromaticum (Clove) Essential Oil" which is also cited in the paper.
My comments follow:
The name of the bacterium Salmonella enterica serotype Enteritidis is not spelled correctly, so correct it
I do not see a description of the method of determining the MIC iMBCte as you determined the MIC50 and MIC 90.
I don't understand how you determined MALDI_TOF MS Biotyper mass spectrometry biofilm? You referred to an earlier article but I don't understand how you picked up the bacteria in the biofilm with just a swab? The biofilm is too tightly bound to the surface to be separated from the surface with just a swab. What changes in the protein profile of the treated biofilm are you talking about? Can it be determined from the peaks on which EO proteins act?
Author Response
Reviewer #3
In the manuscript "Chemical Composition, Antioxidant, In Vitro and In Situ Antimicrobial, Antibiofilm and Anti-Insect Activity of Cedar atlantica Essential Oil" the authors examined the biological effects of Cedar atlantica EO. The work is very similar to their previous work "Chemical Composition, In Vitro and In Situ Antimicrobial and Antibiofilm Activities of Syzygium aromaticum (Clove) Essential Oil" which is also cited in the paper.
My comments follow:
We would like to thank the review for the valuable comments.
Point 1: The name of the bacterium Salmonella enterica serotype Enteritidis is not spelled correctly, so correct it.
Response: It was corrected.
Point 2: I do not see a description of the method of determining the MIC iMBCte as you determined the MIC50 and MIC 90.
Response: We are very sorry it was mistake and we corrected it.
Point 3: I don't understand how you determined MALDI_TOF MS Biotyper mass spectrometry biofilm? You referred to an earlier article, but I don't understand how you picked up the bacteria in the biofilm with just a swab? The biofilm is too tightly bound to the surface to be separated from the surface with just a swab. What changes in the protein profile of the treated biofilm are you talking about? Can it be determined from the peaks on which EO proteins act?
Response: It was previously described: Different stages of the biofilm development during growth and the application of MALDI-TOF MS Biotyper for the evaluation of molecular differences in biofilms grown on different surfaces were evaluated. Growing planktonic cells were used for biofilms. The experiment was performed in 50 mL polypropylene tubes. Each tube contained 20 mL MHB, a glass slide and a wooden toothpick. Prior to the inoculation, cells were incubated in MHB at 37 °C for 24 h. Ten μL of suspension was added into the polypropylene tubes. For the evaluation of the effect of coriander essential oil, the MHB was enriched with 0.1% of essential oil. The prepared samples were incubated at 37 °C in a shaker with a 45 ° inclination at 170 rpm. The biofilm and planktonic cell samples were collected on days 3, 5, 7, 9, 12, and 14 of cultivation. Biofilm samples from the microscope slide and toothpick were taken with a sterile cotton swab and transferred directly on the MALDI-TOF MS Biotyper plate. Planktonic cells were collected by centrifugation of 300 µl of the medium previously collected to an Eppendorf tube at 3000 g for 3 min. The supernatant was resuspended in 25 µl of ultrapure water and the washing procedure was performed three times. Next, 1 μL of the suspension was applied to the MALDI-TOF MS Biotyper plate (Bruker Daltonics, Germany) in duplicate.
The samples were covered with 1 µl of an acyano-4-hydroxy-cinnamic acid matrix (10 mg/mL) and dried at room temperature. After crystallization, the samples were processed with MALDI-TOF MicroFlex (Bruker Daltonics, Germany) linear and positive mode for the range of m / z 2000 – 2000. The spectra were obtained by an automatic analysis and the same sample similarities were used to generate the standard global spectrum (MSP), 40 spectra of MALDI Biotyper 3.0 software were used for the demonstration of all the phases of biofilm development (Bruker Daltonics). MSP spectra were processed by a standard procedure using the MALDI Biotyper dendrogram method and the most representative sample of each spectrum was selected based on common characteristics. The spectra obtained in the present study were compared with the FlexAnalysis 3.0 database (Bruker Daltonics). From the spectra generated by MALDI Biotyper 3.0, 11 MSP were grouped into dendrograms using Euclidean distances.
By comparing the spectra, it is possible to determine from peaks on which EO proteins act, but this was not the aim of our work. The aim was to find out how the structure of biofilms changes according to the surface by recognition with planktonic cells.
Reviewer 4 Report
The paper entitled Chemical Composition, Antioxidant, In Vitro and In Situ Antimicrobial, Antibiofilm and Anti-Insect Activity of Cedar atlantica Essential Oil within the scope of the journal Plants. The Authors have obtained significant and important results that may be of interest to readers and other researchers involved into the issue of utilization of EOs mainly in agriculture and food production as well as their use as a part of organic production and food preservation.
Nevertheless, several problems/doubts should be solved before the manuscript is suitable to be published.
Abstract:
1. L23-24: In this sentence, not all used methods are indicated. Please indicate all used methods or modify the phrase.
2. L 29-31: Please check the statement: “The lowest values of MIC were determined against M. luteus in gram positive and negative bacteria as well as against C. krusei in yeasts and fungi”. I suggest rephrasing it to: “The lowest values of MIC were determined against gram-positive M. luteus (7.46 µL/mL) and against yeast C. krusei (9.46 µL/mL).”
Introduction:
3. L 49: “Vapor” phase. Please use the small letter for “vapor”.
4. L 60-65: Please revise these statements. From these statements, it is not clear about the previous studies of chemical composition and biological activity of EO from plant genus Cedrus in general and from Cedrus atlantica in particular. Has there been any previous research on the phytochemical composition of essential oil from Cedrus atlantica? At the end of the chapter the authors could underline the novelty/innovations of their research with respect to previous papers on the same matter. Please also mention why there is a need for such type of study.
Results:
5. Subsection 2.2, L 85 and Subsection 4.3, L 377: Please indicate the units of TEAC. The concentration of a Trolox standard solution can be expressed in µmol, mg, etc. The TEAC units depend on this and can be different.
6. Table 2: I recommend adding one column in the Table and indicating gram-positive, gram-negative, biofilm-forming bacteria, yeast, and fungi.
7. Tables 3-5: The names of bacteria should be changed to M. luteus and S. marcescens as they were previously mentioned in the text.
8. Footnotes of Table 2: Are the data presented in this Table as mean ± SD? Please add the appropriate information to the footnotes.
9. Footnotes of Tables 2-8: Please add the number of replicates. For example, Means ± standard deviation (n=3).
Discussion:
10. L 261-272: This is the repetition of results. In this part of the Discussion, it will be better to discuss what chemical compounds determined by authors in CEO what substances are responsible for the antibacterial effect? Why Cedar atlantica essential oil is more effective against gram-positive bacteria as compared to gram-negative bacteria?
11. L 292-302: Please additionally discuss the possible reasons and mechanisms of stimulating effect (probacterial activity and profungal activity) of some concentrations of CEO. The results are especially interesting with celery and S. marcescens (Table 5). However, the authors did not discuss it at all.
Material and Methods:
12. L401-402: Why authors did use 0.1% DMSO as negative control? Was CEO for MIC determination diluted in 0.1% DMSO? Please add the description of MIC determination procedure to MM Section.
13. L 399-402: Please add the concentration of antibiotics used in experiments. This information should be also added to the footnote of Table 2.
14. Subsection 4.5: By the description of in situ antibacterial analysis the authors did not mention bread and celery and by the description of the analysis of the antifungal effect the authors did not mention carrot and celery. Please add the missing information.
15. L440: Please add the information on where and how were insects (Pyrrhocoris apterus) obtained.
16. Subsection 4.8: Please add the description of used statistical methods to the Materials and Methods Section more in detail. What statistical test was used to evaluate the significance of the difference in mean values?
17. The number of replicates/repetitions of experiments should be added to the description of each analysis.
Conclusions
18. L 455-458: The sentence is confusing. Please rephrase.
19. L 467-468: “Similar CEO can be very good anti-insect activity against P. apterus” is not English correct. Please re-write.
Author Response
Reviewer #4
The paper entitled Chemical Composition, Antioxidant, In Vitro and In Situ Antimicrobial, Antibiofilm and Anti-Insect Activity of Cedar atlantica Essential Oil within the scope of the journal Plants. The Authors have obtained significant and important results that may be of interest to readers and other researchers involved into the issue of utilization of EOs mainly in agriculture and food production as well as their use as a part of organic production and food preservation.
Nevertheless, several problems/doubts should be solved before the manuscript is suitable to be published.
We would like to thank the review for the valuable comments.
Point 1: Abstract: L23-24: In this sentence, not all used methods are indicated. Please indicate all used methods or modify the phrase.
Response: It was modified.
Point 2: L 29-31: Please check the statement: “The lowest values of MIC were determined against M. luteus in gram positive and negative bacteria as well as against C. krusei in yeasts and fungi”. I suggest rephrasing it to: “The lowest values of MIC were determined against gram-positive M. luteus (7.46 µL/mL) and against yeast C. krusei (9.46 µL/mL).”
Response: It was corrected.
Point 4: Introduction: L 49: “Vapor” phase. Please use the small letter for “vapor”.
Response: It was corrected.
Point 5: L 60-65: Please revise these statements. From these statements, it is not clear about the previous studies of chemical composition and biological activity of EO from plant genus Cedrus in general and from Cedrus atlantica in particular. Has there been any previous research on the phytochemical composition of essential oil from Cedrus atlantica? At the end of the chapter the authors could underline the novelty/innovations of their research with respect to previous papers on the same matter. Please also mention why there is a need for such type of study.
Response: It was mentioned generally. In the end of Introduction was mentioned novelty of the study.
Point 6: Results: Subsection 2.2, L 85 and Subsection 4.3, L 377: Please indicate the units of TEAC. The concentration of a Trolox standard solution can be expressed in µmol, mg, etc. The TEAC units depend on this and can be different.
Response: Concentration and units were added.
Point 7: Table 2: I recommend adding one column in the Table and indicating gram-positive, gram-negative, biofilm-forming bacteria, yeast, and fungi.
Response: It was added.
Point 8: Tables 3-5: The names of bacteria should be changed to M. luteus and S. marcescens as they were previously mentioned in the text.
Response: One of reviewer mentioned full name of bacteria in tables for better understanding of readers.
Point 9: Footnotes of Table 2: Are the data presented in this Table as mean ± SD? Please add the appropriate information to the footnotes.
Response: It was added.
Point 10: Footnotes of Tables 2-8: Please add the number of replicates. For example, Means ± standard deviation (n=3).
Point 11: Discussion: L 261-272: This is the repetition of results. In this part of the Discussion, it will be better to discuss what chemical compounds determined by authors in CEO what substances are responsible for the antibacterial effect? Why Cedar atlantica essential oil is more effective against gram-positive bacteria as compared to gram-negative bacteria?
Response: Thank you for valuable comments but in our study were not tested substances of essential oils, only EO. We add some study of chemical compounds and their effect. From our results and from different authors is it proven that EOs have a stronger effect per gram, but often it also depends on the type of microorganism.
Point 12: L 292-302: Please additionally discuss the possible reasons and mechanisms of stimulating effect (probacterial activity and profungal activity) of some concentrations of CEO. The results are especially interesting with celery and S. marcescens (Table 5). However, the authors did not discuss it at all.
Response: It was add some study in vapor phase with bread, celery and carrot with different essential oil.
Point 13: Material and Methods: 12. L401-402: Why authors did use 0.1% DMSO as negative control? Was CEO for MIC determination diluted in 0.1% DMSO? Please add the description of MIC determination procedure to MM Section.
Response: Yes, EO was diluted with 0.1 % DMSO. Sorry for this mistake, minimal inhibition concentration was added to MM.
Point 14: L 399-402: Please add the concentration of antibiotics used in experiments. This information should be also added to the footnote of Table 2.
Response: It was added.
Point 15: Subsection 4.5: By the description of in situ antibacterial analysis the authors did not mention bread and celery and by the description of the analysis of the antifungal effect the authors did not mention carrot and celery. Please add the missing information.
Response: This part was re-described.
Point 16: L440: Please add the information on where and how were insects (Pyrrhocoris apterus) obtained.
Response: It was added.
Point 17: Subsection 4.8: Please add the description of used statistical methods to the Materials and Methods Section more in detail. What statistical test was used to evaluate the significance of the difference in mean values?
Response: Statistics was evaluated only for MIC.
Point 18: The number of replicates/repetitions of experiments should be added to the description of each analysis.
Response: It was added.
Point 19: Conclusions: L 455-458: The sentence is confusing. Please rephrase.
Response: It was corrected.
Point 20: L 467-468: “Similar CEO can be very good anti-insect activity against P. apterus” is not English correct. Please re-write.
Response: It was corrected.
Round 2
Reviewer 1 Report
Dear Editor,
The authors made all necessary changes and additions and I believe that the paper is now ready for publication. I congratulate all authors for making great efforts for preparing and revising stages of the paper.
Author Response
Reviewer #1
The authors made all necessary changes and additions and I believe that the paper is now ready for publication. I congratulate all authors for making great efforts for preparing and revising stages of the paper.
Response: Thank you for positive feedback.
Reviewer 3 Report
The authors have adopted all the suggestions and I have no further comments.
Author Response
Reviewer #3
The authors have adopted all the suggestions and I have no further comments.
Response: Thank you for positive feedback.
Reviewer 4 Report
The authors improved the manuscript well. I am satisfied with most of the authors' responses to comments, but I have still some remarks:
1) I am grateful to the authors for taking into account my remark regarding the addition of information on the number of replicates. However, I suggest adding this information not to the Title of the Tables, but in footnotes (Tables 3-8). For example, Means ± standard deviation (n=3). In Table 2 it could be put to the phrase: (mm, mean ± SD, n=3).
2) I do not agree with the authors’ comment on point 17. Please check it again:
Point 17: Subsection 4.8: Please add the description of used statistical methods to the Materials and Methods Section more in detail. What statistical test was used to evaluate the significance of the difference in mean values?
Response: Statistics was evaluated only for MIC.
As we can see in Tables 3-8, the authors did designate the statistically different means with different small letters. However, there is no description of what test was used for evaluating the significance of the difference in mean values. Did the authors use Tukey's HSD (honestly significant difference) test? or LSD (least significant difference)? or Student–Newman–Keuls (SNK) method? or Duncan's new multiple range test? or Kruskal–Wallis test? or something else? Was it ANOVA performed? Was the data checked for normality and homogeneity? Some statistical tests can be used only for parametric data. This information is lacking in MM Section as well as in the Tables’ footnote. Please add the appropriate information.
3) I do not agree with the authors’ comment on point 12. Please check it again:
Point 12: L 292-302: Please additionally discuss the possible reasons and mechanisms of stimulating effect (probacterial activity and profungal activity) of some concentrations of CEO. The results are especially interesting with celery and S. marcescens (Table 5). However, the authors did not discuss it at all.
Response: It was add some study in vapor phase with bread, celery and carrot with different essential oil.
In this comment, I suggested the authors discuss not antimicrobial activity and inhibitory action of CEO, but its probacterial activity/ profungal activity (stimulating effect).
The authors write: “Our results showed that CEO had a strong antibacterial effectiveness against the growth of M. luteus and S. marcescens on bread, carrot and celery model.” However, in Table 5 we can see that the result showed that CEO had a strong probacterial effectiveness against the growth S. marcescens on celery model. What is the reason for this effect?
Author Response
Reviewer #4
We thank the opponent very much for excellent and careful review of this article. The authors are grateful for all the comments. We highly value and appreciate the work of the reviewer.
All relevant changes are marked with yellow colour.
The authors improved the manuscript well. I am satisfied with most of the authors' responses to comments, but I have still some remarks:
Point 1: I am grateful to the authors for taking into account my remark regarding the addition of information on the number of replicates. However, I suggest adding this information not to the Title of the Tables, but in footnotes (Tables 3-8). For example, Means ± standard deviation (n=3). In Table 2 it could be put to the phrase: (mm, mean ± SD, n=3).
Response: It was corrected.
Point 2: I do not agree with the authors’ comment on point 17. Please check it again:
Point 17: Subsection 4.8: Please add the description of used statistical methods to the Materials and Methods Section more in detail. What statistical test was used to evaluate the significance of the difference in mean values?
As we can see in Tables 3-8, the authors did designate the statistically different means with different small letters. However, there is no description of what test was used for evaluating the significance of the difference in mean values. Did the authors use Tukey's HSD (honestly significant difference) test? or LSD (least significant difference)? or Student–Newman–Keuls (SNK) method? or Duncan's new multiple range test? or Kruskal–Wallis test? or something else? Was it ANOVA performed? Was the data checked for normality and homogeneity? Some statistical tests can be used only for parametric data. This information is lacking in MM Section as well as in the Tables’ footnote. Please add the appropriate information.
Response: It was changed.
Point 3: I do not agree with the authors’ comment on point 12. Please check it again:
Point 12: 292-302: Please additionally discuss the possible reasons and mechanisms of stimulating effect (probacterial activity and profungal activity) of some concentrations of CEO. The results are especially interesting with celery and S. marcescens (Table 5). However, the authors did not discuss it at all.
In this comment, I suggested the authors discuss not antimicrobial activity and inhibitory action of CEO, but its probacterial activity/ profungal activity (stimulating effect).
The authors write: “Our results showed that CEO had a strong antibacterial effectiveness against the growth of M. luteus and S. marcescens on bread, carrot and celery model.” However, in Table 5 we can see that the result showed that CEO had a strong probacterial effectiveness against the growth S. marcescens on celery model. What is the reason for this effect?
Response: It was corrected. As these are biological experiments, we often cannot clearly explain any of the results. In our case, the results in in situ tests were not clear. The reason may be the created environment for the growth of microorganisms, the concentration of EO used, the microorganism used as well as the representation of individual components in EO. Similar results were found for other essential oils.
We found out in our study that CEO was strongly effective against the growth of M. luteus and S. marcescens on carrot and celery model which is not in line with another study where no significant differences in inhibition between Gram - positive and Gram - negative bacteria in the vaporphase tests were found. Furthermore, E. faecalis (Gram-positive) was the most resistant to clove oil, whereas S. choleraesuis (Gramnegative) was the least inhibited by cinnamon oil. We also noticed some profungal effects on bread and celary model against the growth of some Penicillium species. The effect was found to be cidal (inhibition percentage remains constant with time after removal of the antimicrobial atmosphere) for all of the organisms except A. flavus. In this case, there was a reduction in percentage inhibition during the second week observed. It remained constant for the rest of the testing period after this reduction. There was a prolongation of the tests over 35 days in order to the control of the ability of EOs to provide the protection. There was observed no growth for any of the microorganisms except A. flaVus which confirm the static hypothesis for the latter organism. It is also interesting that estragol had no relevance in the antimicrobial effects of the tested EOs in the vapor phase which proves that the basil seems to be totally ineffective.